# Thermal and Structural Characterization of a Titanium Carbide/Carbon Composite for Nuclear Applications

**DOI:** 10.3390/ma15238358

**Published:** 2022-11-24

**Authors:** Michele Ballan, Stefano Corradetti, Mattia Manzolaro, Giovanni Meneghetti, Alberto Andrighetto

**Affiliations:** 1National Institute of Nuclear Physics—Legnaro National Laboratories (INFN-LNL), Viale dell’Università 2, Legnaro, 35020 Padova, Italy; 2Department of Industrial Engineering, University of Padova, Via Venezia 1, 35131 Padova, Italy

**Keywords:** titanium carbide, thermal characterization, structural characterization

## Abstract

In the framework of ISOL (isotope separation on-line) facilities, porous carbides are among the most employed target materials for the production of radioactive ion beams for research. As foreseen by the ISOL technique, a production target is impinged by an energetic particle beam, inducing nuclear reactions from such an interaction. The resulting radionuclides are subsequently released, thanks to the high target working temperature (1600–2000 °C); ionized; and extracted into a beam. Since the target microstructure and porosity play a fundamental role in the radionuclide release efficiency, custom-made target materials are often specifically produced, resulting in unknown thermal and structural properties. Considering that such targets might undergo intense thermal stresses during operation, a thermal and structural characterization is necessary to avoid target failure under irradiation. In the presented work, a custom-made porous titanium carbide that was specifically designed for application as an ISOL target was produced and characterized. The thermal characterization was focused on the evaluation of the material emissivity and thermal conductivity in the 600–1400 °C temperature range. For the estimation of a reference material tensile stress limit, the virtual thermoelastic parameter approach was adopted. In particular, for the aforementioned temperature range, an emissivity between 0.7 and 0.8 was measured, whereas a thermal conductivity between 8 and 10 W/mK was estimated.

## 1. Introduction

Since its invention in 1951, the isotope separation on-line (ISOL) technique has been employed worldwide for the production of radioactive ion beams (RIBs) in large international research facilities, such as ISOLDE [1], ISAC [2], HRIBF [3], SPIRAL [4], and ALTO [5], and the growing interest on RIBs is promoting the installation of new ISOL facilities. In such a framework, the Italian National Institute for Nuclear Physics (INFN) is constructing a new ISOL facility at Legnaro National Laboratories (LNL) called the Selective Production of Exotic Species (SPES) [6].

As foreseen by the ISOL technique, a charged particle beam interacts with the atoms within a production target, inducing nuclear reactions such as spallation, fragmentation, or fission. The resulting reaction products are subsequently released, thanks to the high target working temperature (2000 °C); ionized; and extracted into a beam. Carbides have been widely used as solid target materials for the production and online release of radioisotopes thanks to their refractoriness, which allows them to sustain an operational temperature between 1600 °C and 2000 °C in high vacuum, and their capability to be shaped as powders, disks, pellets, and sintered rods [7]. The working temperature, the microstructure, and the morphological properties of the targets, such as open porosity, play a crucial role in the efficient release of the produced nuclides.

The typical SPES target is a multifoil porous uranium carbide (UC_x_) target composed of seven thin 40 mm diameter disks that are axially spaced to optimize the dissipation by thermal radiation of the power deposited by a 40 MeV 200 μA proton beam, and it is used for the production of neutron-rich RIBs in the 80–160 amu mass range [8]. In the case of other specific nuclei of interest, e.g., neutron-deficient radionuclides, other refractory carbides such as silicon carbide (SiC), lanthanum carbide (LaC_x_), boron carbide (B4C), and titanium carbide (TiC) are being adopted as target materials, maintaining the same multi-disk-shaped architecture proposed for UC_x_. In particular, SiC was tested to produce aluminum RIBs at Oak Ridge National Laboratories (ORNL) in similar conditions [9] and is planned as the first material to be employed for the commissioning of the SPES facility, as it implies a lower radiological risk compared to UC_x_.

Among the other aforementioned materials, titanium carbide has been extensively tested as an ISOL target material at both CERN-ISOLDE [10,11] and TRIUMF-ISAC [12] and is currently under investigation for the production of scandium RIBs at both SPES and the ISOL@MYRRHA facility in the Belgian SCK•CEN institute. In particular, since the latter will exploit protons in the energy range of 0–100 MeV, the same 40 mm diameter multidisk target architecture is being investigated [13].

Titanium carbide is a synthetic, superhard refractory material that is commonly used for manufacturing metal-working tools, protective coatings, and carbide steel [14]. The high density, which is a requirement for such applications, represents a drawback in the case of ISOL target materials, as it affects the release of the produced nuclides. Indeed, as highlighted in recent studies, high porosity, especially open interconnected porosity, and a high specific surface area, together with a low grain size, have a positive effect on the achievable isotope yields [15,16]. For this reason, the TiC employed for ISOL applications is often custom-made, enhancing the desired microstructure and porosity. As innovative production techniques are explored, such custom porous TiC types, which also include sol–gel-based or nanostructured materials, exhibit properties that can be widely different from the well-characterized commercially available samples. In addition, porous carbides for ISOL applications are widely used as composites containing carbon in different forms, with the aim of improving the diffusion rate and the high-temperature stability of the porosity and microstructure [17].

Different from other ISOL facilities, where a highly energetic particle beam passes through a target, depositing only a small fraction of its power, the SPES target components are affected by a remarkably high power density, as the 40 MeV proton beam is completely stopped within the target. This extreme working condition entails thermal stresses within the target disks that could eventually lead to their failure if their stress limit is reached. As a result, the characterization of the target material’s thermostructural behavior is a crucial aspect for both target design and the safe operation of the facility [18].

Additionally, the use of composite materials can improve specific material thermophysical properties, such as the thermal conductivity and the tensile fracture strength, and decrease the coefficient of thermal expansion. Such effects are highly desirable in view of ISOL target applications since they reduce the target material failure due to the thermal stresses that can arise during operation. A similar efficient tailoring of material properties has been achieved for industrial as well as light-shielding applications thanks to the use of nanoparticle-based polymer composites [19,20,21].

In this work, taking the strategy adopted for the thermal and mechanical characterization of a commercial dense SiC as reference, the production procedure of custom porous composite TiC/C disks is described with the experimental measurement of their spectral emissivity and the estimation of their thermal conductivity. The stress limit of this custom material was estimated according to the virtual thermoelastic parameter (VTP) approach that was theorized in previous works [18]. This method allows the evaluation of the target’s structural design, even if thermoelastic material properties, such as Poisson’s ratio, Young’s modulus, and the coefficient of thermal expansion, are unknown. It is relevant to highlight that the aforementioned procedure was successfully applied for the first time to a custom-produced composite material characterized by a high porosity.

The operational temperature limit for this material was not studied in detail in the presented work since it was accurately determined in previous studies [11].

## 2. Materials and Methods

### 2.1. Materials

High-purity titanium oxide (≥99%, Honeywell, Merck, Milan, Italy) and high-purity graphite powder (≥99.99%, particle size <45 μm, Aldrich Chemistry, Merck, Milan, Italy) were used as precursors for the sample production.

### 2.2. Production of the TiC samples

The procedure selected for the TiC sample production was the classical carbothermal reaction with titanium dioxide (TiO_2_) as a precursor and graphite powder as a carbon source.

In particular, a crystalline graphite powder was used, for which a detailed XRD spectrum was available from previous studies [22]. Regarding the TiO_2_ powder, no information was provided by the supplier about its crystalline form (rutile or anatase).

An excess of carbon, necessary for the occurrence of the reaction, was introduced following the stoichiometric ratios in Reaction (1):(1)$${\mathrm{TiO}}_{2}+5C\;\to \mathrm{TiC}+2C+2\mathrm{CO},$$

The procedure for the production of the green pellets involved the grinding and mixing of the TiO_2_ powder with the graphite in a ball mill at 400 rpm for 45 min. The powders were subsequently transferred into an agate mortar, where a phenol-formaldehyde resin in an acetone solution was added as a binder and manually incorporated into the powder mix, corresponding to 5% wt. of resin in the mix. After acetone evaporation, the obtained mixture was pressed in the form of 40 mm disk-shaped pellets under a uniaxial pressure of 300 MPa for 10 min.

Subsequently, the green pellets were loaded horizontally onto a graphite tray and inserted into a graphite furnace, where they were treated in high vacuum (10^−5^ mbar) at 1900 °C with a heating rate of 2 °C/min, a dwell time of 24 h, and a cooling rate of 2 °C/min. Besides being necessary for the occurrence of Reaction (1), the thermal treatment also had the aim of favoring the partial sintering of the freshly formed TiC.

### 2.3. Dimensional and Morphological Characterization

The dimensional and morphological characterizations were focused on the evaluation of the macro- and microstructure of the produced samples. The dimensional characterization involved both a visual inspection of the sample to evaluate its shape regularity and the measurement of its diameter and thickness with a digital caliper. Such observations were performed both before and after the carboreduction thermal treatment in order to quantify the apparent volume reduction and assess any eventual deformation that occurred during the sintering process.

The sample’s mass before and after the thermal treatment was also measured to quantitatively verify the weight loss due to the release of carbon oxides and the degradation of the organic binders. The apparent density (ρ_bulk_) of the resulting samples was then calculated as the ratio of the mass and the apparent volume after sintering. With such data, it was possible to calculate the total porosity (P) of the produced TiC disks using the following equation, considering a theoretical density (ρ_th_) of 3.38 g/cm^3^ [23], which was calculated considering the volume fractions of TiC and C in the theoretical TiC + 2C composition:(2)$$P=1-\frac{\rho_{\mathrm{bulk}}}{\rho_{\mathrm{th}}},$$

After sintering, the flat surfaces of the samples were observed with a scanning electron microscope (SEM-TESCAN model VEGA 3) to visually assess the resulting microstructure.

### 2.4. High-Temperature Thermal Characterization

The aim of the high-temperature thermal characterization was to evaluate the material properties that define the target temperature field during operation at a temperature as close as possible to 2000 °C. In steady-state conditions in high vacuum, only radiative and conductive thermal fluxes concur in the heat exchange phenomenon. For this reason, the corresponding target material properties, namely emissivity (ε) and thermal conductivity (k), were sufficient to define the target temperature field.

For the estimation of ε and k at a high temperature, two specific experimental procedures were developed in previous works based on a dedicated experimental setup (Figure 1) [18,24]. This device consisted of a vacuum chamber, capable of operating at vacuum levels around 10^−6^ mbar, containing a graphite Ohmic heater. The latter was opportunely shaped to generate a circular hot spot of approximately 20 mm diameter at its center, whose temperature was uniform and could be increased up to 2000 °C by the stepwise increment of the heating current. A sample disk with a diameter between 30 and 40 mm was positioned and centered with the heater hot spot using three tungsten support rods, preventing direct contact with the graphite resistor. As a consequence, only radiative fluxes concurred with the sample heating since the contact with the support rods was punctual. The resulting sample temperature distribution typically exhibited a radial pattern with a hot spot at the center. In addition to the modulation of the heating current, the extent of the temperature gradient between the sample center and the periphery could be varied by adjusting the distance between the graphite resistor and the tested disk. The vacuum chamber was equipped with a borosilicate viewport positioned in correspondence with the top surface of the sample disk, allowing both visual observation and noncontact temperature measurements. In particular, the latter were performed using an infrared ratio pyrometer (IRCON^®^ modline 5 R, Fluke Process Instrument, Everett, WA, USA) characterized by wavelength working bands of 0.75–1.05 μm and 1.0–1.1 μm in two-color mode and 1.0–1.1 μm in one-color mode and output temperature ranges of 600–1400 °C for the 5R-1410000 model and 1000–3000 °C for the 5R-3015000 model. When the instrument operated in one-color mode, accurate sample temperature measurements required spectral emissivity at ~1 μm ε_1 μm_ of the surface of interest, which was often unknown, especially in the case of custom-manufactured samples. Conversely, when the pyrometer worked in two-color mode, the sample temperature was directly derived from the ratio of the infrared emission in the two working wavelength bands, without knowing the corresponding spectral emissivity value. The potential wavelength-dependent variation in the emission for each type of sample material was taken into account by opportunely setting the e-slope parameter, as recommended by the manufacturer.

The adopted pyrometer control software allowed temperature acquisition as well as the measurement of the 1 μm spectral emissivity of the sample surface through an automatic routine. In particular, the sample’s punctual temperature was first acquired in two-color mode. Then, the pyrometer switched to one-color mode, and the emissivity setting was automatically varied until the temperature readout matched the two-color mode acquisition, thus obtaining the ε_1 μm_ of the measured temperature.

The typical test procedure involved a stepwise increase in the Ohmic heater current. When steady-state conditions were reached for each current step, temperature and emissivity were measured at the sample’s center and periphery.

The procedure for the estimation of the sample’s thermal conductivity, k, was based on both the aforementioned experimental measurements and on a consolidated finite element (FE) model that accurately reproduced the adopted test bench. This model, implemented in ANSYS^®^, allowed the analysis of both the electrical problem and the radiative/conductive heat exchange processes (convection could be neglected thanks to the high-vacuum working conditions). In steady-state conditions, provided that the thermal and electrical characteristics of the graphite heater were well known, the model required the definitions of the temperature-dependent ε(T) and k(T) to accurately simulate the sample temperature field. The presented experimental routine allowed for the measurement of the normal spectral emissivity, ε_1 μm_(T), as a function of the acquired temperature. Concerning the thermal radiation problem, the model adopted the assumption of grey diffuse surfaces. Therefore, the required sample radiative property was the total hemispherical emissivity, ε_tot_(T). As highlighted in previous studies, the directional dependence of the radiation emission is generally negligible [25], and carbides are often reasonably assumed to be grey bodies, omitting the potential infrared wavelength influence. Consequently, the measured ε_1 μm_(T) could be considered a valid estimation of the ε_tot_(T) required for the ANSYS^®^ model. With such hypotheses, it follows that k(T) provides the only unknown sample property. As proposed in previous works, the temperature-dependent thermal conductivity can be expressed within the FE model in the form of a second-degree polynomial, such as k(T) = C_0_ + C_1_T + C_2_T^2^. Therefore, the estimation of k(T) consisted of the numerical determination of the triplet of C_i_ coefficients of the polynomial for which the computed temperature distribution was consistent with the experimentally measured data. In ANSYS^®^, this was handled as a typical optimization problem, where the C_i_ coefficients were the optimization variables and the objective function was the residual function J(**C**), expressed as Equation (3):(3)$$J\left(C\right)=\sum_{i=1}^{N}{\left[T_{C_{{\mathrm{FE}}_{i}}}\left(C\right)-T_{C_{{\mathrm{EXP}}_{i}}}\right]}^{2}+{\left[T_{P_{{\mathrm{FE}}_{i}}}\left(C\right)-T_{P_{{\mathrm{EXP}}_{i}}}\right]}^{2}\;P=1-\frac{\rho_{\mathrm{bulk}}}{\rho_{\mathrm{th}}},$$
where **C** is the Ci optimization variable vector, N corresponds to the number of heating current steps applied for the experimental measurements, $T_{C_{{\mathrm{FE}}_{i}}}\;\left(C\right)$ and $T_{P_{{\mathrm{FE}}_{i}}}\;\left(C\right)$ are the temperatures computed using the FE model at the center and periphery of the sample, respectively, and $T_{C_{{\mathrm{EXP}}_{i}}}$ and $T_{P_{{\mathrm{EXP}}_{i}}}$ are the corresponding experimental data. The optimization process was performed by using the ANSYS^®^ APDL optimization tool to obtain the set of C_i_ that minimize the residual function J(**C**).

### 2.5. Estimation of the Stress Limit through the Virtual Thermoelastic Parameter (VTP) Approach

The characterization of the stress limit of a material is a fundamental step for the evaluation of its suitability as a target, especially in the case of SPES. Indeed, when the proton beam impinges the target disks, a typical radial temperature pattern is generated. Such a temperature distribution can result in intense thermal stresses within the target disks, which can cause a fracture if the material stress limit, σ_lim_, is reached.

The same experimental test bench used for the thermal characterization could be used to perform fracture tests by thermal stresses on the sample disks. This procedure, described in detail in previous works [18], involved the gradual intensification of such stresses by slowly increasing the heating current on the graphite heater up to the fracture of the sample. In the case of ceramic samples, a brittle fracture was expected. The output of this testing procedure was the heating current intensity, I_fract_, for which the sample failure was observed. At this point, the steady-state sample temperature distribution at the failure moment {T_fract_} was calculated using the same FE model adopted for the thermal conductibility estimation. In this case, the sample’s thermal properties, ε_tot_(T) and k(T), and the facture current intensity, I_fract_, were provided as input data. Subsequently, the FE model was used to perform a thermostructural analysis to compute the stress field that caused the sample failure {σ_fract_} once the proper thermoelastic properties were introduced. In steady-state conditions, for isotropic linear elastic materials, such as ceramics, the Young’s modulus, E; the Poisson’s ratio, ν; and the coefficient of thermal expansion, α, were the properties required to compute {σ} starting from {T}. The radial pattern of {T}, characterized by a sample center–periphery temperature gradient with a peak value at the center, produced a typical stress distribution with compression at the center of the disk, where both the radial and circumferential stress components, σ_r_ and σ_θ_, respectively, were negative stresses (the axial component, σ_z_, was negligible since {σ} had an evident planar distribution). These stress components were not dangerous for the structural integrity of the disk. Conversely, at the sample periphery, the stress component σ_θ_ exhibited its maximum positive value and corresponded to the maximum tensile first principal stress (σ_I_) within the sample. As ceramic materials are generally characterized by a weak resistance to tension, it was reasonable to ascribe the sample failure to the maximum positive value of the σ_θfract_ component of the computed stress field {σ_fract_} at the disk periphery. This value could be identified as the critical tensile stress, σ_C_, of the considered specimen. The computed fracture stress data from the different samples were subsequently analyzed trough the Weibull statistical approach, which was applicable in the case of the brittle fracture of ceramic materials, as seen in the ASTM practice [26]. The material σ_lim_ in the tested temperature range was determined as the stress associated with a survival probability of 99.99%.

In the case of custom-produced ceramic materials, the application of the aforementioned procedure for the σ_lim_ estimation is not trivial, as the thermoelastic material properties are often unknown and difficult to determine experimentally. For this reason, the virtual thermoelastic parameter (VTP) approach was theorized and presented in previous works, where it was applied the sake of comparison to fully characterized commercial silicon carbides. According to this approach, arbitrary temperature-independent values for the parameters E*, ν*, and α* can be assumed for the sake of the calculation of a virtual stress field {σ*}. In this way, it was possible to estimate a σ*_lim_ of the material trough a Weibull analysis. With this reasonable assumption, if the same values of E*, ν*, and α* were used in the design phase of a target, it was possible to compare the computed maximum first principal stress, σ*_I MAX_, with the estimated σ*_lim_ for the structural verification. Table 1 summarizes the adopted virtual thermoelastic parameters E*, ν*, and α*, which were extracted from available literature data at room temperature for other kinds of titanium carbides.

## 3. Results

### 3.1. Production of the TiC Samples

After cooling, mechanically stable disk-shaped samples were extracted from the furnace. A total of 46 specimens was successfully produced.

### 3.2. Dimensional and Morphological Characterization

Table 2 summarizes the average sample thickness and diameter both before and after the thermal treatment. The presented results are averaged over a total of 46 samples and are reported together with the calculated standard deviation, reasonably assuming a Gaussian distribution. Whereas the disk thickness was essentially unaffected by the thermal treatment, a diameter reduction of approximately 10% was observed.

A comparison of the disk mass before and after the thermal treatment is reported in Table 3. The highlighted mass reduction is attributable to the degradation/outgassing of the phenolic resin, together with the release of carbon oxides produced by the carbothermal reaction. The theoretical weight loss associated with Reaction (1) is about 41% (also considering the small mass loss relative to the phenolic resin decomposition). The correspondence between the theoretical and experimental data seems to suggest that Reaction (1) was complete at the end of the thermal treatment. As desired, a high porosity was achieved for all samples, with an average value of 58%.

The flat top surface of the TiC specimens was observed using an SEM microscope. Figure 2 reports the typical observed superficial pattern: the material was composed of two phases, the formed TiC matrix and a dispersion of residual unreacted graphite clusters, as expected by the designed stoichiometry of TiC + 2C. XRD or EDS analyses for the verification of the possible presence of a residual phase of TiO_2_ were not performed since similar previous studies highlighted that a thermal treatment in vacuum at a temperature above 1700 °C ensures total oxide degradation [23,30,31]. All specimens exhibited crack-shaped pores that were homogenously distributed on the sample surface, confirming the expected high porosity.

### 3.3. High-Temperature Thermal Characterization

For the high-temperature thermal characterization, three different samples were tested as built, with no additional thermal treatment or surface-finishing postprocessing. Each sample underwent four subsequent heating cycles in which the emissivity and temperature data for the estimation of thermal conductivity were collected simultaneously. The samples were tested with different heater–sample gaps. In particular, sample 1 was positioned at a distance of 1.22 mm, sample 2 was positioned at a distance of 1.85 mm, and sample 3 was positioned at a distance of 3.11 mm.

Figure 3 reports the 1 μm spectral normal emissivity data measured for the three samples in the 600–1400 °C temperature range. Consistent data were collected for all samples, exhibiting a slightly increasing emissivity trend between 0.7 and 0.8.

Figure 4 summarizes the estimated thermal conductivity, k(T), trends for the three aforementioned samples with their respective 95% confidence bounds. The resulting k(T) generally ranged between 7 and 11 W/m °C in the 600–1400 °C temperature range, and all samples exhibited consistent data within their 95% confidence bounds (CB). In particular, the estimations for samples 1 and 2 resulted in quasi-identical slightly increasing trends, whereas in the case of sample 3, k(T) had a quasi-constant tendency in the considered temperature range.

Figure 5 reports the comparison between the experimentally measured temperature values at the center and periphery of sample 1 and the corresponding numerical data, which were calculated considering the reported estimations for ε(T) and k(T). The experimental measurements and the FEM calculations generally exhibited consistent results.

### 3.4. Estimation of the Stress Limit through the Virtual Thermoelastic Parameter (VTP) Approach

Stress limit estimation through the virtual thermoelastic parameter (VTP) approach is a method based on the statistical Weibull analysis. Therefore, it requires the testing of a sufficiently large amount of samples. In the case of this study, 24 TiC specimens were tested with the aim of inducing intense thermal stresses capable of leading to sample failure. Three different outcomes occurred for such tests, as reported in Figure 6, where three representative samples are displayed.

In particular, for 16 of the tested samples, the complete and instantaneous fracture of the specimen was observed, with a diametric or quasi-diametric crack propagating for the whole disk diameter (Figure 6a), producing two or three fragments. Six other disks exhibited the sudden occurrence of a crack that did not propagate up to the complete fracture of the sample (Figure 6b). Two TiC specimens did not exhibit evident signs of failure since no visible cracking was observed, even if they were heated with the maximum available heating power (Figure 6c). All disks with incomplete or non-evident failure were subsequently retested with another heating cycle to further investigate the crack propagation/occurrence, respectively, but no assessable modifications were observed.

The heating current intensity, I_fract_, for which the sample failure occurs was identified when the fragments of the broken sample fell from the tungsten support rods and a sudden discontinuity in the pyrometer signal was reported. Consequently, a clear identification of I_fract_ was not feasible in the case of the six incompletely cracked specimens since the pyrometer temperature measurement on the sample surface exhibited a continuous signal.

Table 4 reports the data for the 16 samples that resulted in complete failure in terms of the recorded I_fract_ and the last measured center temperature before breakage. It is relevant to highlight that such specimens were tested with different sample–heater distances, resulting in different temperature distributions at equal heating powers. Additionally, as highlighted in Table 4, failure occurred in most cases at a temperature lower than the maximum temperature level reached during the thermal characterization tests. Indeed, such experimental activity was performed on samples that did not exhibit evident signs of failure, similar to the specimen shown in Figure 6c.

For each of the 16 specimens, the virtual stress field at the fracture {σ*_fract_} was calculated with the VTP approach, and the maximum positive value of the σ*_θfract_ component at the sample periphery was extracted, which corresponded to the critical tensile stress, σ*_C_, of the sample. These data were subsequently analyzed with the statistical Weibull approach according to the ASTM standard practice [26]. Figure 7 reports the resulting data distribution, with P_f_ being the failure probability, which was calculated as follows:(4)$$P_{f,i}=\frac{i-0.5}{N},$$
where i is the ith sample and N the total specimen number.

Table 5 summarizes the Weibull distribution parameters with 90% confidence bounds and the material virtual stress limit, σ*_lim_. This value was computed from the Weibull distribution, taking a failure probability of 0.01% as a reference, which corresponded to a survival probability of 99.99%.

## 4. Discussion

In the presented study, the production and characterization of a porous titanium carbide suitable for ISOL applications were performed. The selected production technique, which involved the carbothermal reduction of TiO_2_ in the presence of a carbon source, was already consolidated in previous works [23,30]. However, in this study, this procedure was successfully used for the manufacture of approximately 40 mm diameter thin disks for the first time. This sample dimension was selected to guarantee the full compatibility of the manufactured specimens with the peculiar SPES ISOL target architecture [32]. The sample morphological and dimensional analysis confirmed the high repeatability of the production process since all samples exhibited very similar dimensions. Additionally, a high porosity above 50% was achieved, and SEM observations highlighted the presence of interconnected crack-shaped pores. On one hand, this microstructure is expected to promote the release of the produced nuclides when this material is used as an ISOL production target. For some specific radionuclides, such as scandium isotopes, previous studies highlighted that a high porosity may not be sufficient to promote an efficient nuclide release from titanium carbide [16,33]. The development of nanostructured target materials could overcome such limitations. On the other hand, the presence of cracks has a detrimental effect on the mechanical resistance of the material that can result in the material being highly prone to failure in the presence of the typical thermal stresses that arise during operation at a high temperature. This phenomenon is however mitigated by the presence of a consistent residual graphite phase that normally has a beneficial effect on the material’s mechanical resistance.

The thermal characterization was aimed at the estimation of the target emissivity and thermal conductivity, which are the properties that define the radiative and conductive heat fluxes on the target in steady-state conditions. In the presented study, the 600–1400 °C temperature range was explored, with 600 °C being the minimum measurable value of the available infrared pyrometers and 1400 °C being the maximum temperature reached during the experimental campaign. The 1 μm spectral emissivity data collected for all three samples exhibited a consistent quasi-constant behavior between 0.7 and 0.8. This range is typical for this kind of carbide and was observed in previous works for similar materials [18]. Furthermore, the consistency of the data collected for the three different samples confirmed the repeatability of the production technique.

The estimation of the material’s thermal conductivity was successfully performed with a procedure that was developed and consolidated for dense ceramic materials based on the employment of an ANSYS^®^ finite element model that accurately reproduced the experimental test bench. On average, the studied titanium carbide exhibited a thermal conductivity in the range of 8–10 W/m °C, with a slightly increasing trend as a function of the temperature. This rising behavior is ascribable to the presence of the dispersed graphite phase and was also observed for other composites with a graphite dispersion within a carbide matrix [24,34,35]. In addition, a similar range of thermal conductivity was also measured for a high-porosity silicon carbide in previous works [36].

It is interesting to highlight that the computed confidence bounds were wider the farther is the sample was from the heater. Indeed, the most accurate estimations were performed in the case of sample 1, which was tested with a specimen–heater gap of 1.22 mm. This effect can be related to the simplifications introduced in the FE model, where the assumption of diffuse surfaces was adopted for the radiative heat transfer modeling. In the case of greater sample–heater distances, the directional dependence of the radiative heat flux intensity, which was neglected in the FE model, might have a slight effect on the specimen temperature distribution. However, the estimations performed on sample 3, tested with a 3.11 mm gap, were reasonably consistent with the other two samples. Therefore, they can be considered acceptably accurate. 

Another crucial step for the thermal characterization of an ISOL target material candidate is the identification of its limit temperature, namely the maximum operational temperature in the vacuum range 10^−5^–10^−6^ mbar [18]. Normally, this parameter is estimated by evaluating the weight loss of samples during a long-term heating process at a given uniform temperature. In the case of the considered material, several studies in the literature reported the range 1900 ÷ 2000 °C as the operational temperature for TiC with a disperse graphite phase [10,11,12,33]. Since the reported data referred to a material with a similar composition produced with the same approach, the experimental determination of the limit temperature was not necessary.

Once the thermal properties were characterized, the estimation of the tensile stress limit with the virtual thermoelastic parameter approach was possible. As a custom material, in the case of the titanium carbide presented in this work, its thermoelastic properties are unknown. Therefore, the VTP approach represented a valid tool for the calculation of a referential stress limit that could be adopted in the target design phase for the identification of its optimal working conditions. It is relevant to highlight that the VTP approach, which was theorized and validated with different grades of commercial silicon carbides in previous works [18], was applied for the first time to a custom porous material in this study.

Of the 24 tested samples, only 16 were used for the Weibull analysis, as a complete specimen failure was only observed in those cases, and it was therefore possible to compute σ*_C_ with VTP. Indeed, six samples did not exhibit any detectable evidence of failure. Therefore, their critical tensile stress values were not reached during the tests. Since all samples were tested with the same thermal treatment, it was reasonable to assume that the intact titanium carbide disks were characterized by higher critical tensile stress values than the specimens that shattered into two or more fragments. Additionally, two samples presented visible cracks but did not shatter into fragments, and neither crack propagated with subsequent heating cycles. Indeed, since the cracks stopped at the central region, the occurrence of compressive thermal stresses blocked the crack propagation during the test. For the application as a target at SPES, this failure scenario does not represent an issue. Indeed, if a target disk does not shatter into fragments, it can still operate for the production of RIBs. Furthermore, the presence of a crack significantly reduces the extent of the tensile stresses at the target’s periphery, decreasing its failure likelihood. For such reasons, the presented Weibull analysis, performed neglecting eight such samples, led to a more conservative estimation of the material’s σ*_lim_, as samples with higher σ*_C_ values were not considered in the calculations. At this point, it is relevant to clarify that the computed σ*_lim_ does not represent the absolute stress limit of the considered material, but it is a reference value that the target designer can adopt for design validations or operation condition evaluations as long as the thermal stress field is computed with the same virtual thermoplastic parameters that were adopted in this study.

## 5. Conclusions

The presented work, carried out in the framework of the research and development of high-power targets for nuclear applications, was aimed at the characterization of a custom porous titanium carbide that is suitable as an ISOL target. Titanium carbide with a disperse graphite phase was produced, and a porosity higher than 50% was reached. The thermal characterization allowed the evaluation of the sample emissivity and thermal conductivity in the 600–1400 °C temperature range, adopting a consolidated method that involved the employment of a specifically designed test bench that was reproduced by a finite element model. An emissivity between 0.7 and 0.8 was measured, and a thermal conductivity in the range of 8–10 W/mK was estimated.

For the structural verification of the target disks, the virtual thermoelastic parameter approach was used, and a reference tensile stress limit was estimated. This information, together with the evaluated thermal properties and the literature data regarding the limit temperature, will be fundamental for the assessment of the target operation conditions in both the design and commissioning phases in the context of the SPES project. Indeed, such information could be used to predict, with an FEM simulation, the effect of the impinging primary beam on the target and to identify the acceptable beam characteristics in terms of beam size, intensity, and energy.

As an interesting future development, the effects of different densities and microstructures can be evaluated, with the aim of enhancing the material suitability as an ISOL target. On one hand, the maximization of the thermal conductivity is highly desirable, as a more homogeneous temperature distribution leads to less intense thermal stresses, reducing the risk of target failure. This effect can be achieved by either increasing the material density or considering other shapes for the graphite disperse phase, for example, radially oriented fibers of opportune length. On the other hand, as the porosity has a crucial role in the target’s capability to release the produced isotopes, an excessive increase in the target’s density should be avoided.

## Figures and Tables

**Figure 1 materials-15-08358-f001:**
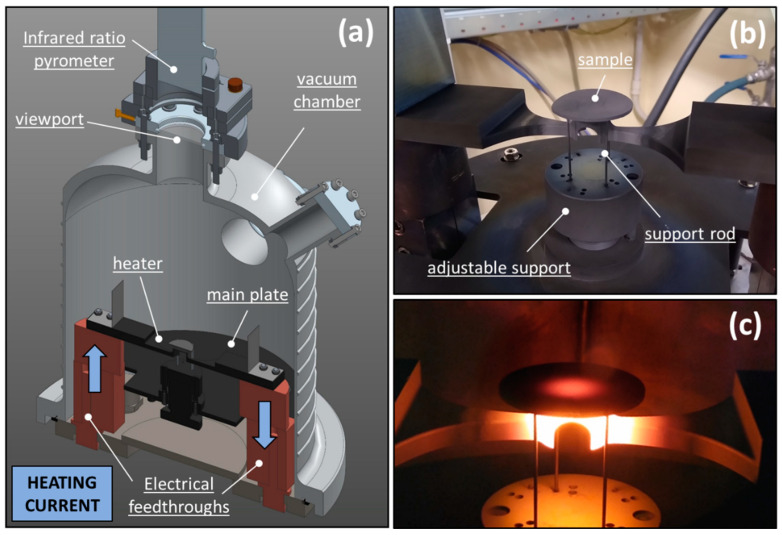
The experimental test bench adopted for the measurement of the sample emissivity and temperature data necessary for the thermal conductivity estimation: general CAD view (**a**), detailed view of the sample and heater (**b**), and pictures of the components at high temperature during a test with a TiC specimen (**c**).

**Figure 2 materials-15-08358-f002:**
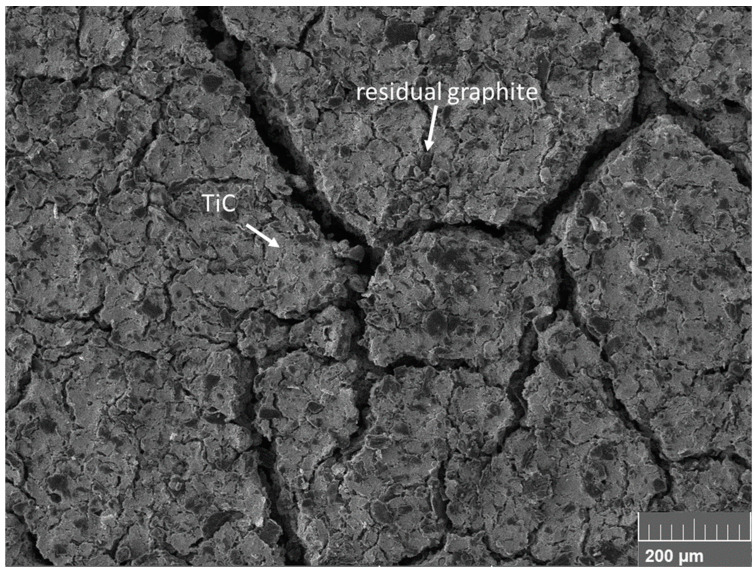
SEM picture of the flat surface of a TiC specimen.

**Figure 3 materials-15-08358-f003:**
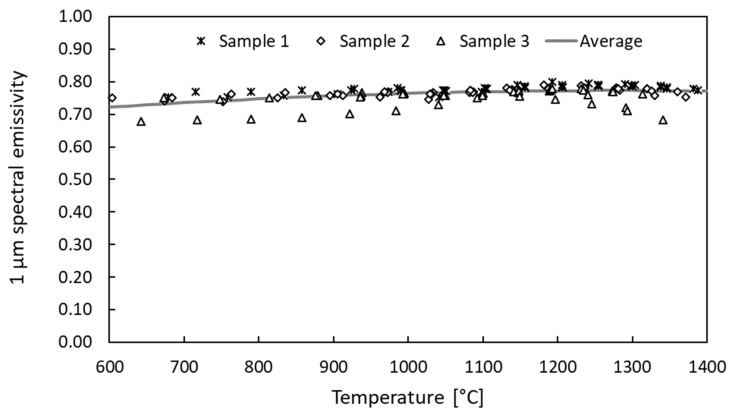
The 1 μm spectral emissivity data collected for three different samples.

**Figure 4 materials-15-08358-f004:**
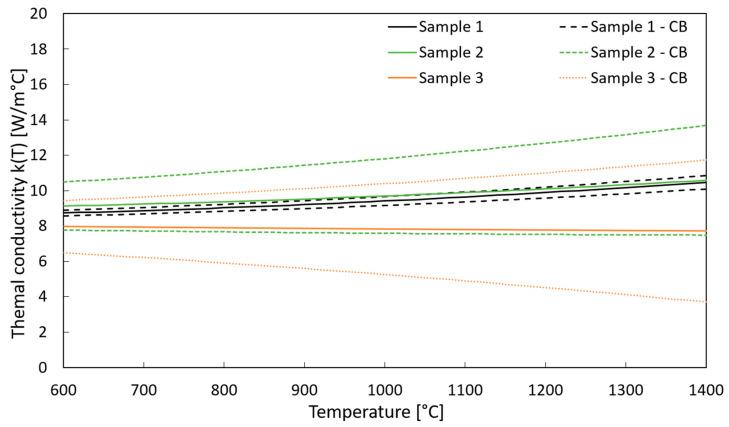
Estimated thermal conductivity for three different samples with the corresponding 95% confidence bounds (CBs).

**Figure 5 materials-15-08358-f005:**
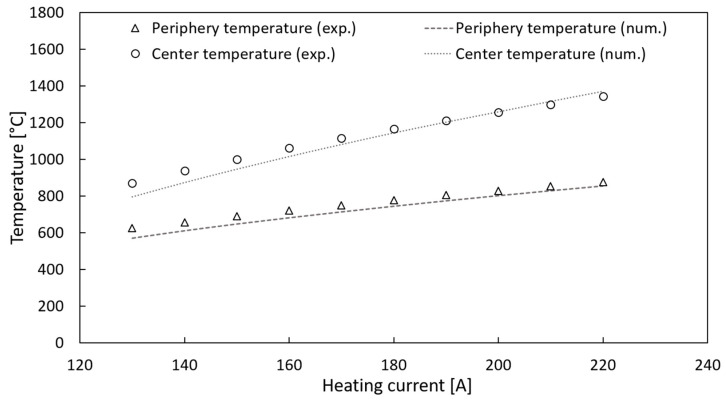
Comparison between the experimental and numerically calculated temperature values at the center and periphery of sample 1.

**Figure 6 materials-15-08358-f006:**
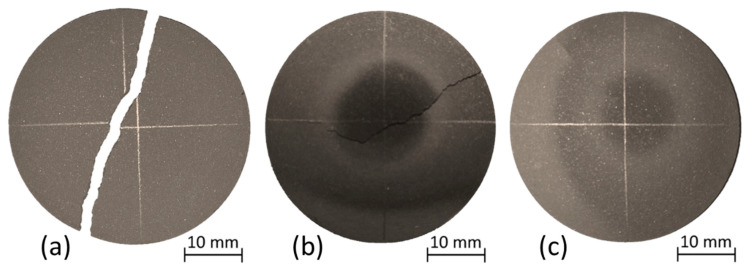
The three outcomes of the performed thermal stress failure tests: complete failure (**a**), incomplete crack propagation (**b**), and no evident cracking (**c**).

**Figure 7 materials-15-08358-f007:**
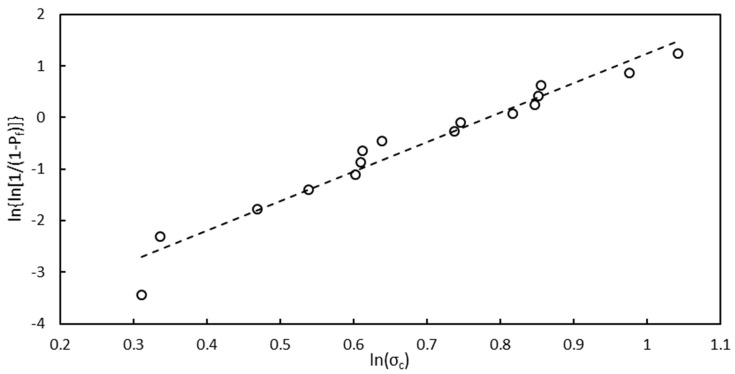
Plot of the Weibull distribution for the 16 tested samples.

**Table 1 materials-15-08358-t001:** Elastic properties from the literature, which were adopted for the calculation of the stress field according to the virtual thermoelastic parameter approach.

Young’s Modulus (E*)	Poisson’s Ratio (ν*)	Coefficient of Thermal Expansion (α*)
3220 (MPa) [27]	0.191 [28]	7.4∙10^−6^ (1/°C) [29]

**Table 2 materials-15-08358-t002:** Summary of the average dimensions of the produced TiC samples before and after the thermal treatment.

	Before Thermal Treatment	After Thermal Treatment
Diameter (mm)	40.5 ± 0.2	36.6 ± 0.7
Thickness (mm)	1.56 ± 0.05	1.57 ± 0.06

**Table 3 materials-15-08358-t003:** The average mass of the produced TiC samples before and after the thermal treatment and the achieved average porosity.

Average Mass before Thermal Treatment (g)	Average Mass after Thermal Treatment (g)	Average Mass Loss	Average Porosity
3.97 ± 0.07	2.34 ± 0.12	41 ± 3%	58 ± 3%

**Table 4 materials-15-08358-t004:** Data recorded for the 16 samples for which a complete failure was observed.

Sample Number	Sample Thickness (mm)	Sample Diameter (mm)	Heater-Sample Distance (mm)	Fracture Current Intensity (I_fract_) (A)	Center Temperature at the Fracture (°C)
1	1.53	35.95	3.11	165	752
2	1.51	36.06	3.11	150	892
3	1.58	35.95	3.11	160	999
4	1.61	36.34	4.77	160	966
5	1.61	36.08	3.11	170	1032
6	1.63	36.24	3.11	170	1040
7	1.62	35.87	3.11	190	1131
8	1.57	36.00	1.22	160	1369
9	1.55	36.07	1.22	120	814
10	1.63	36.17	1.85	140	917
11	1.62	36.24	1.85	140	874
12	1.58	36.07	3.11	140	874
13	1.62	36.02	3.11	160	960
14	1.55	35.81	1.22	150	951
15	1.62	36.17	1.22	120	795
16	1.65	36.12	1.85	140	851

**Table 5 materials-15-08358-t005:** The Weibull distribution parameters (with 90% confidence bounds) and the referential stress limit calculated with the VTP approach.

$\hat{m}$ (/) (Lower Bound)	$\hat{m}$ (/) (Average)	$\hat{m}$ (/) (Upper Bound)	$\hat{\sigma }$_θ_ (MPa) (Lower Bound)	$\hat{\sigma }$_θ_ (MPa) (Average)	$\hat{\sigma }$_θ_ (MPa) (Upper Bound)	σ*_LIMIT_ (MPa)	Temperature Range (°C)
3.72	5.72	7.38	2.03	2.21	2.40	0.44	700 ÷ 1400

## Data Availability

Not applicable.

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
