# Peer review of "Thermal and Structural Characterization of a Titanium Carbide/Carbon Composite for Nuclear Applications"

_materials, 2022, doi:10.3390/ma15238358_

Round 1
Reviewer 1 Report
This paper studies TiC as ISOL target for nuclear applications. The experimental design is reasonable, with a few minor points needing revisions. See my comments below:
1. Abstract: “ (1600÷2000°C)” should be “ (1600-2000°C)”.
2. Materials/Methods: After the reaction to produce TiC, are there impurities of C and TiO2 remaining in the powder? If so, how much and how is it measured? If not, how do impurities affect the remaining material tests?
3. Results: what is the purpose of Figure 2? What information does it provide to the audience?
4. Results: Figure 7, please provide the scale bars. Also, I’m not sure the current way of classifying wafers based on shown in Figure 7 is the best way. Is the figure shown representative of the whole sample? Also, did authors conduct any TEM study at the failure sites?
Reviewer 2 Report
Dear Authors,
In this manuscript, the authors have synthesized a new type of nanocomposites by multistage solution method and investigated various important properties that made this paper interesting. However, the manuscript needs several corrections before publication. The summarized comments and queries regarding this manuscript are presented point by point as follows:
Major revisions:
Comment 1: In the abstract and conclusion highlights the resulting properties and specific applications of the synthesized samples more delicately. In the introduction, specify the device applications with more relevant citations and improve the introduction based on the following references, Radiation Effects, and Defects in Solids Volume 174, Issue 5-6, Pages 480 - 4933 June 2019; Results in Physics Open Access Volume 13 June 2019 Article number 102264; Khan et al, 2 - Classification and properties of nanoparticles, 2022, Pages 15-54, ISBN 9780128242728). Therefore, revise the introduction, abstract, and conclusions carefully.
Comment 2: The originality and novelty of the research work are not properly displayed in the introduction section. It is better to mention the crystalline form of the starting materials in the experiment. In Table 1 why commas are used instead of dots? Why division symbols are used among digital data in many places? Correct these.
Comment 3: Texts and resolution of Figures 4-6 and 8 are not published, please improve. Tables formatting and symbols presentation can be superior.
Minor revisions:
Comment 1: Several typos, symbols presentation, spacing among words and units, commas, and full stops missing as well as several language corrections are recommended throughout the manuscript.
The manuscript is impressive for the science community, and readers that can be published after amending all the raised issues in this review process.
